# PLA/Graphene/MWCNT Composites with Improved Electrical and Thermal Properties Suitable for FDM 3D Printing Applications

**Evgeni Ivanov** [1,2,*] **, Rumiana Kotsilkova** [1] **, Hesheng Xia** [3] **, Yinghong Chen** [3] **, Ricardo K. Donato** [4] **, Katarzyna Donato** [4] **, Anna Paula Godoy** [4] **, Rosa Di Maio** [5] **, Clara Silvestre** [5] **, Sossio Cimmino** [5] **and Verislav Angelov** [1]

[1] Open Laboratory on Experimental Micro and Nano Mechanics (OLEM), Institute of Mechanics, Bulgarian Academy of Sciences, Acad. G. Bonchev Str. Block 4, 1113 Sofia, Bulgaria; kotsilkova@yahoo.com (R.K.); verislav@abv.bg (V.A.)
[2] Research and Development of Nanomaterials and Nanotechnologies (NanoTech Lab Ltd.), Acad. G. Bonchev Str. Block 4, 1113 Sofia, Bulgaria
[3] State Key Laboratory of Polymer Materials Engineering, Polymer Research Institute of Sichuan University, Chengdu 610065, China; xiahs@scu.edu.cn (H.X.); johnchen@scu.edu.cn (Y.C.)
[4] MackGraphe—Graphene and Nanomaterials Research Center, Mackenzie University, Rua da Consolação 896, São Paulo 01302-907, Brazil; ricardo.donato@mackenzie.br (R.K.D.); zawada.kat@gmail.com (K.D.); apaulasgodoy@gmail.com (A.P.G.)
[5] Istituto per i Polimeri, Compositi e Biomateriali (IPCB), Consiglio Nazionale delle Ricerche (CNR), Via Campi Flegrei 34 Olivetti, 80078 Pozzuoli (NA), Italy; rosadimaio1988@libero.it (R.D.M.); clara.silvestre@ipcb.cnr.it (C.S.); sossio.cimmino@ipcb.cnr.it (S.C.)
[*] Correspondence: ivanov_evgeni@yahoo.com; Tel.: +359-2979-6481

**Abstract:** In this study, the structure, electrical and thermal properties of ten polymer compositions based on polylactic acid (PLA), low-cost industrial graphene nanoplates (GNP) and multi-walled carbon nanotubes (MWCNT) in mono-filler PLA/MWCNT and PLA/GNP systems with 0–6 wt.% filler content were investigated. Filler dispersion was further improved by combining these two carbon nanofillers with different geometric shapes and aspect ratios in hybrid bi-filler nanocomposites. Scanning electron microscopy (SEM), transmission electron microscopy (TEM) and Raman spectroscopy exhibited uniform dispersion of nanoparticles in a polymer matrix. The obtained results have shown that for the mono-filler systems with MWCNT or GNP, the electrical conductivity increased with decades. Moreover, a small synergistic effect was observed in the GNP/MWCNT/PLA bi-filler hybrid composites when combining GNP and CNT at a ratio of 3% GNP/3% CNT and 1.5% GNP:4.5% CNT, showing higher electrical conductivity with respect to the systems incorporating individual CNTs and GNPs at the same overall filler concentration. This improvement was attributed to the interaction between CNTs and GNPs limiting GNP aggregation and bridging adjacent graphene platelets thus, forming a more efficient network. Thermal conductivity increases with higher filler content; this effect was more pronounced for the mono-filler composites based on PLA and GNP due to the ability of graphene to better transfer the heat. Morphological analysis carried out by electron microscopy (SEM, TEM) and Raman indicated that the nanocomposites present smaller and more homogeneous filler aggregates. The well-dispersed nanofillers also lead to a microstructure which is able to better enhance the electron and heat transfer and maximize the electrical and thermal properties. The obtained composites are suitable for the production of a multifunctional filament with improved electrical and thermal properties for different fused deposition modelling (FDM) 3D printing applications and also present a low production cost, which could potentially increase the competitiveness of this promising market niche.

**Keywords:** biodegradable polymers; graphene; carbon nanotubes; nanocomposites; electrical and thermal properties

## 1. Introduction

In the last few years, 3D printing has emerged as a leading manufacturing technology all over the world for a variety of applications. The success of 3D printing depends on fine-tuned materials to the needs of each application. There are many thermoplastic and thermosetting polymeric materials that can be selected for 3D printing depending on the application [1]. Polylactic acid (PLA)—a thermoplastic polymer derived from natural sources, completely biodegradable, and bio-absorbable—is considered a green alternative to petrochemical commodity plastics. It is highly demanded due to versatility in applications such as packaging, pharmaceuticals, textiles, engineering, chemical industries, automotive composites, and biomedical and tissue engineering; lately material for 3D printing has been promising [2,3]. However, its mechanical performance as well as electrical and thermal properties must be improved in order to expand the application fields. The incorporation of nanofillers is a common approach to attain this goal.

Carbon-based nanomaterials, offer the potential to combine PLA properties with several of their unique features, such as high mechanical strength, electrical conductivity, thermal stability and bioactivity. Carbon nanotubes (CNT) and graphene are state of the art and very promising carbonaceous materials. Their high specific surface area allows low loadings in order to tune polymer key properties concerning mechanical, thermal, electrical, and biological performance. CNT have exceptional mechanical properties, aspect ratio, electrical and thermal conductivities, and chemical stability. These characteristics make them excellent candidates for the creation of multifunctional materials either in the field of polymer composites or other application [4–13]. Based on the unique properties of graphene, the graphene-based polymer composites are expected to offer superior mechanical properties, enhanced electrical and thermal conductivity, improved dimensional stability, higher resistance to microcracking, and increased barrier properties above the matrix polymer [14–18]. Recently, graphene and functionalized graphene were utilized in composites containing different fillers, and the combination of fillers have shown synergy effect in terms of the mechanical properties, thermal and electrical conductivities, and super capacitance [19]. It was found that combining together two nanofillers, such as carbon nanotubes and graphene, leads to the formation of a co-supporting network of both fillers.

Among the different technologies applicable for graphene-polymer products fabrication, the additive manufacturing (e.g., 3D printing) is the most promising since it is generally applicable to different types of polymers and is a versatile and low-cost technology. The development of new conductive polymer nanocomposite materials for 3D printing is highly desirable to achieve better printability, mechanical properties, electrical and thermal conductivity [20–28]. Three methods most frequently used to obtain a dispersion of nanofillers into a polymer matrix in order to produce a filament are: solution mixing, melt blending, and in situ polymerization [29,30]. Melt blending is a very attractive method from an economical point of view; it is environmentally friendly and highly scalable for preparing nanocomposites. This method involves direct addition of the nanofillers into the molten polymer, and thus allows optimization of the dispersion state by adjusting processing parameters such as mixing speed, time and temperature.

The main idea and the novelty in the present work are related with investigation of the synergetic (combined) effects of various mono and bi-filler combinations of low-cost industrial graphene and multi-walled carbon nanotubes on the structure, electrical and thermal conductivity of biodegradable PLA-based nanocomposites. The electrical and thermal properties of as-prepared composites were examined as well as the dispersion of nanofillers in polymer matrix was investigated by transmission electron microscopy (TEM), scanning electron microscopy (SEM) and Raman spectroscopy.

## 2. Materials and Methods

### 2.1. Materials

The polymer matrix used in this study was Ingeo™ Biopolymer PLA-3D850 (Nature Works, Minnetonka, MN, USA) grade developed especially for manufacturing 3D printer monofilament with MFR 7–9 g/10 min (210 °C, 2.16 kg).

The nanofillers used are Industrial Graphene NanoPlates, GNP (supplied by TimeNano, Chengdu, China), with purity, 90 wt.%; number of layers <30; thickness <30 nm; diameter/median size 5–7 µm; aspect ratio: ~230/165, specific surface area, $m^2/g$: 1.42 $m^2/g$, as well as Industrial Grade OH-Functionalized Carbon Nanotubes (multi-walled carbon nanotubes) (MWCNTs) (TimeNano, produced by CVD method) with purity, 95 wt.% OH 2.48% content; size (outer D = 10–30 nm, inner D = 5–10 nm, length = 10–30 µm); aspect ratio: ~1000; specific surface area 110 $m^2/g$; density 2.1 $g/cm^3$; electrical conductivity S = 100 S/cm.

### 2.2. Preparation of Nanocomposites

Robust design pre-planning of compositions, combining different proportions of the two nanocarbon fillers and varying the filler contents from 0 to 6 wt.%, was made. The mono-and bi-filler nanocomposite hybrids were processed using the melt extrusion method trough preparation of masterbatches and further dilution. The mono-filler composites (PLA/MWCNT and PLA/GNP) with 1.5 wt.%, 3 wt.% and 6 wt.% filler contents as well as bi-filler composites (PLA/MWCNT/GNP) with 3 wt.% and 6 wt.% of total filler content (combining GNP and MWCNT in different proportions) were prepared. Table 1 shows the studied compositions.

**Table 1.** The mono-filler and bi-filler composites.

| Composition Code (wt.%) | GNP Content (wt.%) | MWCNT Content (wt.%) | PLA Content (wt.%) | Name |
|---|---|---|---|---|
| PLA | - | - | 100 | referent |
| PLA/1.5% GNP | 1.5 | - | 98.5 | mono-filler |
| PLA/3% GNP | 3 | - | 97 | mono-filler |
| PLA/6% GNP | 6 | - | 94 | mono-filler |
| PLA/1.5% MWCNT | - | 1.5 | 98.5 | mono-filler |
| PLA/3% MWCNT | - | 3 | 97 | mono-filler |
| PLA/6% MWCNT | - | 6 | 94 | mono-filler |
| PLA/1.5% GNP/1.5% MWCNT | 1.5 | 1.5 | 97 | bi-filler |
| PLA/1.5% GNP/4.5% MWCNT | 1.5 | 4.5 | 94 | bi-filler |
| PLA/3% GNP/3% MWCNT | 3 | 3 | 94 | bi-filler |
| PLA/4.5% GNP/1.5% MWCNT | 4.5 | 1.5 | 94 | bi-filler |

### 2.3. Preparation of Test Samples

Specimens of sizes 16 mm diameter and about 3 mm thickness were hot pressed from extruded pellets, polished and then used for electrical and thermal conductivity tests.

### 2.4. Experimental Methods

A picoammeter (Keithley 2400, Keithley Instruments Inc., Beaverton, OR, USA) was used to measure the electrical conductivity of the mono-and bi-filler polymer composites. Before measurements the samples were coated with a silver coating. Silver coating doesn't influence measurements, since it is measured the bulk conductivity while the sample is between two electrodes and silver coating is on both sides of the disc samples (with a diameter of 16 mm and thickness 3 mm). The edges of the samples are thick enough (3 mm) and they are not covered with the silver paint. That is why there is no contact between two silver-painted surfaces. During the electrical measurements, the resistance of the material in ohm was obtained. Electrical conductivity was calculated using the following Equation (1):

$$k = \frac{L}{R \cdot S} \tag{1}$$

where $k$ is electrical conductivity (S/m); $L$ is a thickness of the sample (m); $S$ is the area of the circle ($\pi r^2$).

A Hot Disk 2500 thermal constant analyzer (Hot Disk Inc., Göteborg, Sweden) was used to measure the thermal conductivity of the GNP and MWCNTs polymer composites through a transient plane source method. Before measurements, the samples were polished using an abrasive paper in order to obtain very flat surfaces. The measurements were performed with specimens of sizes 16 mm diameter and about 3 mm thickness by putting the sensor (3 mm diameter) between two similar slabs of material. The sensor supplied a heat pulse of 0.01 W for 40 s to the sample at room temperature and the associated change in temperature was recorded. The thermal conductive parameters including thermal conductivity, thermal diffusivity, and specific heat of the samples were measured.

Bright field transmission electron microscopy (TEM, FEI Company, Hillsboro, OR, USA) analysis was performed by using FEI TECNAI G12 Spirit-Twin (LaB6 source) instrument equipped with a FEI Eagle-4k CCD camera and operating with an acceleration voltage of 120 kV. The analysis was performed on sections obtained at room temperature by using a Leica EM UC6/FC6 ultramicrotome. The sections were placed on 400 mesh copper grids.

Scanning electron microscopy (SEM, FEI Company, Hillsboro, OR, USA) measurements were performed using FEI QUANTA 200F scanning electron microscope at different magnifications. SEM characterizations of the nanocomposite blends were done using brittle fractured composite surface prepared using liquid nitrogen. The sample surface was covered with gold-palladium (Au/Pd) using EMITECH K575X.

The Raman spectra of carbonaceous components were obtained using a Raman Confocal Spectrometer (WITec Alpha 300 R, WITec Wissenschaftliche Instrumente und Technologie GmbH, Ulm, Germany) with a 50× objective lens, grating of 600 grooves/mm and excitation laser wavelength of 532 nm. Laser beam aperture of 1.2 micrometres and measured laser power of 3 mW. Raman spectroscopy is an efficient technique to characterize graphene in terms of quality, structural integrity and number of layers. Thus, single spectra were collected to evaluate the nanocarbon fillers structure/quality within the pelletized filament, collecting an average of 25 spectra per sample at different locations for excluding localized effects. Moreover, Raman spectral imaging (mapping) was performed on hot-pressed films (at 200 °C for 10 min) to selectively evaluate the distribution of GNP and CNT by mapping individually the distribution of the D, G and 2D bands within a longitudinal section of hot-pressed films. Prior to the application of mapping, single spectra of the hot-pressed films were performed to localize the appropriate focus position and detect any effect caused by the hot pressing in the carbonaceous component.

## 3. Results and Discussion

### 3.1. Structure and Morphology

A homogeneous dispersion of MWCNT and GNP in polymeric melts is essential to prepare enhanced filler-based filaments, since heterogeneity can be detrimental to Fused Deposition Modeling technology (FDM), possibly causing blockages at the nozzle and flux instability while printing. Thus, current research has focused on determining the concentration of carbon nanofillers that would surpass the percolation threshold (the transition between an insulating and conductive polymer) while maintaining the parameters for 3D printing. The homogeneity of GNP dispersion directly affects the composite's printability ability and other physical properties. TEM and SEM is a common tool used to distinguish carbon nanofillers from the polymer matrix due to different contrasts and filler specific morphologies. Figure 1a–d presents SEM images of cryo-fractured surfaces of the mono-filler and bi-filler composites containing (a) PLA/6 wt.% MWCNT; (b) PLA/6 wt.% GNP; (c) PLA/1.5% GNP/1.5% MWCNT and (d) PLA/3% GNP/3% MWCNT, respectively. All images are at the same magnification of 20,000×, for comparison. Figure 1a clearly shows the good distribution of carbon

nanotubes (protruding white dots on the cryo-fractured surface) in the polymer matrix at the maximum mono-filler concentration of 6 wt.% and no aggregates were found. Figure 2b presents randomly dispersed single GNP platelets and small aggregates at a maximum mono-filler content of 6 wt.% GNP. Figure 1c,d revealed a distribution of MWCNTs and GNP in the polymer matrix for bi-filler systems, containing the same quantity of each nanofiller PLA/1.5% GNP/1.5% MWCNT and PLA/3% GNP/3% MWCNT. In this case images show randomly distributed single carbon nanotubes, small aggregates of nanotubes and GNP packages.

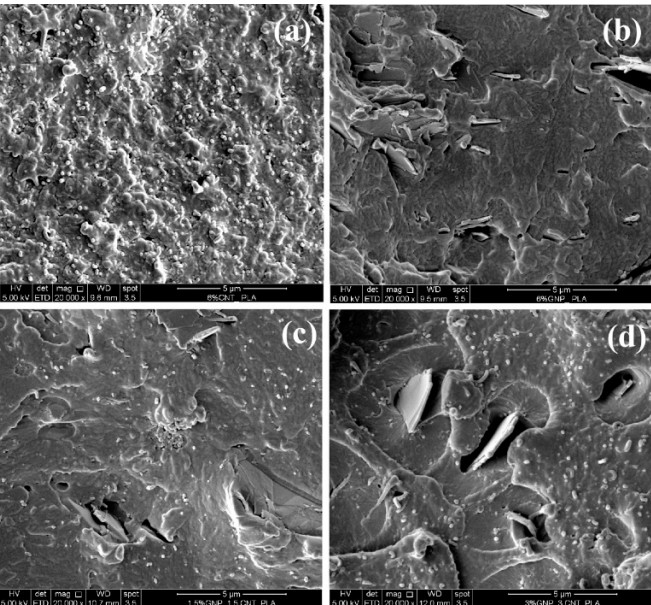

**Figure 1.** Scanning Electron Microscopy (SEM) images of cryo-fractured surfaces of the mono-filler composites, containing (**a**) 6 wt.% multi-walled carbon nanotubes (MWCNT) and (**b**) 6 wt.% of graphene nanoplates (GNP) and bi-filler composites, containing (**c**) 1.5% GNP/1.5% MWCNT and (**d**) 3% GNP/3% MWCNT, respectively at magnification 20,000×.

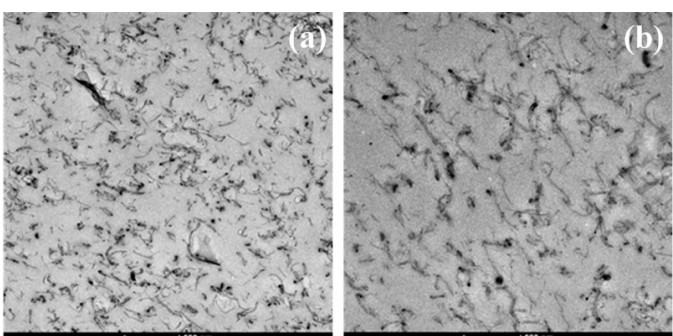

**Figure 2.** Transmission Electron Microscopy (TEM) images of the bi-filler composites containing (**a**) polylactic acid (PLA)/1.5 wt.% GNP/4.5 wt.% MWCNT and (**b**) PLA/3 wt.% GNP/3 wt.% MWCNT, respectively.

Figure 2a,b presents TEM images of the bi-filler composites with a total filler content of 6 wt.% (a) PLA/1.5 wt.% GNP/4.5 wt.% MWCNT and (b) PLA/3 wt.% GNP/3 wt.% MWCNT, respectively. A good bi-filler distribution can be observed at both combined concentrations on Figure 2a,b. Both fillers seem to form an interconnected network, where the single nanotubes, small MWCNT aggregates and GNPs creates an interpenetrated network of particles and small aggregates. Obviously, the combination of the two nanofillers in the PLA matrix led created the possibility of unique synergetic effect for the improvement of the electrical and mechanical properties as described below.

Raman spectroscopy can be used as a quick and unambiguous method to determine the number of graphene layers (flakes or walls in nanotubes) and its structural quality either at a pure state or dispersed in the polymer matrix. Graphene spectrums are characterized by three bands: G, D and 2D. The G band is related to the in-plane vibrational mode that involves $sp^2$ hybridized carbon atoms of graphene sheets; its position is highly sensitive to the number of layers (shifts to lower wavenumber with an increased number of layers). The D band represents defects in the graphene carbon lattice. Additionally, intensity ratios: $I_D/I_G$ and $I_{2D}/I_G$; give an insight of structural disorder degree and quality of graphene, where defect-free graphene ratio is equal 2. The single and sharp second-order Raman band (2D) has been widely used as a simple and efficient way to confirm the presence of single layer graphene. The 2D band of multilayer graphene can be fitted with multiple peaks due to the splitting of the electronic band structure of the multilayer material [31,32]. Figure 3 presents the Raman spectra of neat PLA as well as its composites. Obtained PLA spectrum (Figure 3a) presented very strong bands at around 2900–2990 cm$^{-1}$ (assigned to the CH asymmetric and symmetric stretching region) and strong bands at about 1750 cm$^{-1}$ (assigned to the C=O stretching region), 1450 cm$^{-1}$ (assigned to $CH_3$), 1370 cm$^{-1}$ (assigned to CH deformation and asymmetric bands) and 950 cm$^{-1}$ (helical backbone vibration with $CH_3$ rocking modes) [33]. It was noted that for both GNP concentrations, bands 2900–2990 cm$^{-1}$ decreased with increasing filler content. Sample PLA/1.5 wt.% GNP (Figure 3b) presented more intense D band (at 1356 cm$^{-1}$) than sample PLA/6 wt.% GNP (with D band at 1348 cm$^{-1}$, Figure 3c) which can indicate more structural defects in the graphene lattice, probably due to heat/sheer damage during extrusion. The reduction of structural defects with increasing GNP content (PLA/6 wt.% GNP) could be the result of a lubrication process during the melting process. The small and broad 2D bands for PLA/1.5 wt.% GNP (at 2715 cm$^{-1}$, Figure 3b) and PLA/6 wt.% GNP (at 2719 cm$^{-1}$, Figure 3c) are in accordance to graphite profile. The $I_D/I_G$ and $I_{2D}/I_G$ ratios for PLA/1.5 wt.% GNP (0.66 and 0.37, respectively) revealed a high degree of structural disorder. In the case of PLA/6 wt.% GNP the $I_D/I_G$ and $I_{2D}/I_G$ ratios (0.09 and 0.33, respectively) indicated more surface integrity. Also, the Full Width Half Maximum (FWHM) value must be considered (Table 2). This is a measure of structural disorder, producing a quantitative guide to distinguish up to 5 layers [34]. The PLA/1.5 wt.% GNP nanocomposite showed the FWHM of 96 cm$^{-1}$, which represents a characteristic of multilayered GNP (more than 10 layers). The nanocomposite that presented the lowest FWHM value was PLA/6 wt.% GNP, 79 cm$^{-1}$, and this is also attributed to multi-layered platelets.

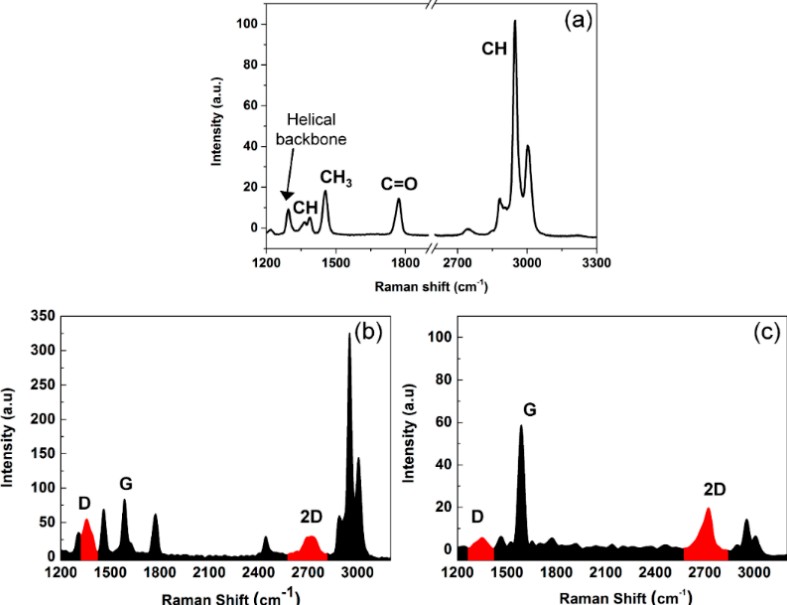

**Figure 3.** *Cont.*

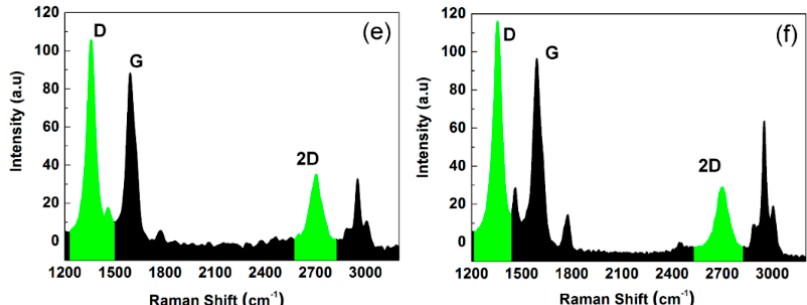

**Figure 3.** Raman spectroscopy results obtained for (**a**) PLA, (**b**) PLA/1.5 wt.% GNP, (**c**) PLA/6 wt.% GNP, (**d**) PLA/1.5 wt.% MWCNT and (**e**) PLA/6 wt.% MWCNT composites.

**Table 2.** Full Width Half Maximum (FWHM) value of mono-filler composites.

| Composition Code | FWHM (cm$^{-1}$) |
| --- | --- |
| PLA/1.5% GNP | 96 |
| PLA/6% GNP | 79 |
| PLA/1.5% MWCNT | 96 |
| PLA/6% MWCNT | 90 |

In the case of PLA/MWCNT composites, the diameter/number of walls of nanotubes, the presence of disorder in sp$^2$-hybridized carbon lattice and the effect of nanotube-nanotube interactions on the vibrational modes have been studied using Raman spectroscopy [35–37]. Herein, broader D and 2D bands represent a higher number of walls, thus higher disorder and larger diameters. According to the literature, the intense G band relates to better nanotube dispersion in the polymer matrix [38]. At both concentrations the G band is very intensive indicating good MWCNTs dispersion and confirming results obtained by SEM (Figure 1a,b).

Figure 4a–c presents the Raman mapping results of PLA/1.5 wt.% GNP/1.5 wt.% MWCNT composites. The D band mapping displays many regions with low filler density (black regions), while the G and 2D band mapping presented more homogeneous and well-dispersed distribution. Since the highest contribution for the D band is given by the CNT, the darker regions could represent a region with less presence of CNT bundles.

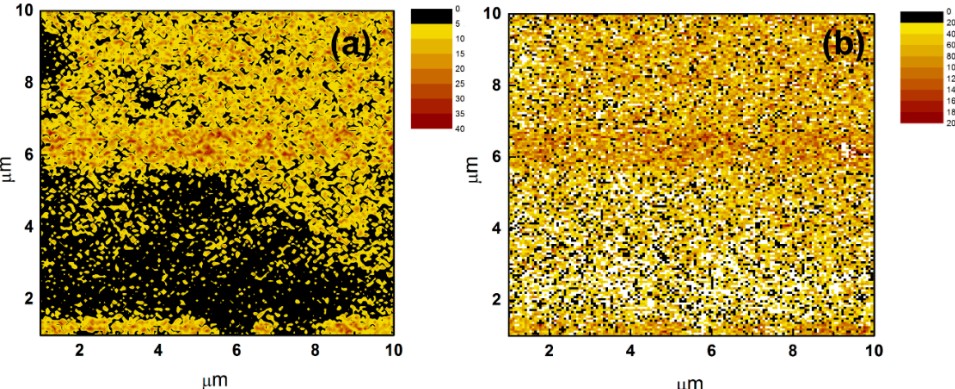

**Figure 4.** *Cont.*

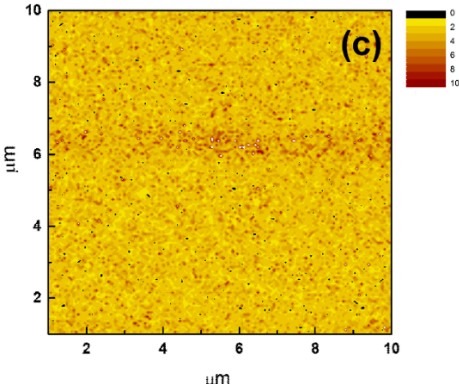

**Figure 4.** Raman mapping of (**a**) D band, (**b**) G band and (**c**) 2D band of the same area for PLA/1.5 wt.% GNP/1.5 wt.% MWCNT composites.

### 3.2. Effect of Filler Concentration and Combinations of Nanofillers on the Electrical Conductivity

Traditional thermoplastic polymers are usually electrical insulators. Polymers containing electrically conductive fillers show interesting electrical properties like semiconductors and metals but without losing the processability of polymers. Typical applications are as antistatic (electrostatic dissipation) materials, electromagnetic interference shielding materials, heaters and sensors. Figure 5 and Table 3 represents electrical conductivity vs. GNP and MWCNT filler content for mono-filler (PLA/GNP and PLA/MWCNT and bi-filler (PLA/GNP/MWCNT) composites.

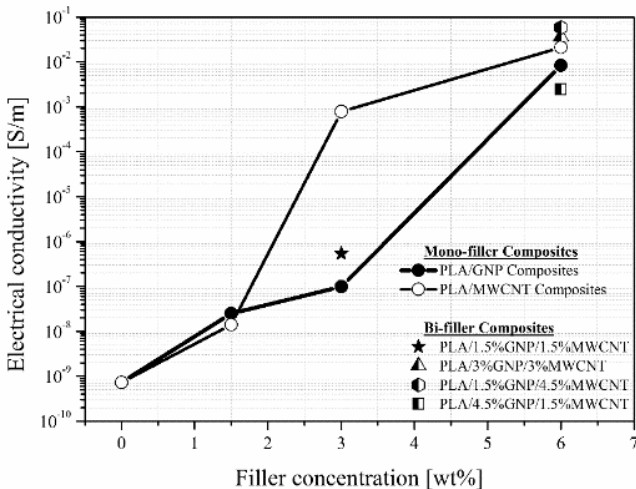

**Figure 5.** Electrical conductivity vs. GNP and MWCNT filler content for mono-filler (PLA/GNP and PLA/MWCNT) and bi-filler (PLA/GNP/MWCNT) composites.

**Table 3.** Electrical and thermal properties of mono-filler and bi-filler composites.

| Composition Code (wt.%) | Electrical Conductivity (S/m) | Thermal Conductivity (W/mK) | Thermal Diffusivity (mm²/s) |
|---|---|---|---|
| PLA | $7.3 \times 10^{-10}$ ($\pm 3.39 \times 10^{-10}$) | 0.205 ($\pm$0.0045) | 0.154 ($\pm$0.0002) |
| PLA/1.5% GNP | $2.51 \times 10^{-8}$ ($\pm 1.48 \times 10^{-9}$) | 0.2695 ($\pm$0.014) | 0.245 ($\pm$0.0001) |
| PLA/3% GNP | $9.9 \times 10^{-8}$ ($2.97 \times 10^{-8}$) | 0.3748 ($\pm$0.0011) | 0.3079 ($\pm$0.0002) |
| PLA/6% GNP | 0.00835 ($\pm$0.00233) | 0.577 ($\pm$0.0151) | 0.485 ($\pm$0.0018) |
| PLA/1.5% MWCNT | $1.4 \times 10^{-8}$ ($\pm 2.83 \times 10^{-9}$) | 0.2275 ($\pm$0.0026) | 0.1887 ($\pm$0.0001) |
| PLA/3% MWCNT | $7.86 \times 10^{-4}$ ($\pm 4.86 \times 10^{-4}$) | 0.2612 ($\pm$0.0026) | 0.1859 ($\pm$0.0001) |
| PLA/6% MWCNT | 0.021 (0.00707) | 0.303 ($\pm$9.9.10-4) | 0.221 ($\pm 8.7 \times 10^{-4}$) |
| PLA/1.5% GNP/1.5% MWCNT | $5.4 \times 10^{-7}$ ($\pm 7.07 \times 10^{-8}$) | 0.3013 ($\pm$0.0071) | 0.2339 ($\pm$0.0002) |
| PLA/1.5% GNP/4.5% MWCNT | 0.0585 ($\pm$0.0318) | 0.3779 ($\pm$0.0031) | 0.2551 ($\pm$0.0001) |
| PLA/3% GNP/3% MWCNT | 0.036 ($\pm$0.0112) | 0.4253 ($\pm$0.0213) | 0.3138 ($\pm$0.002) |
| PLA/4.5% GNP/1.5% MWCNT | 0.00244 ($\pm 8.56 \times 10^{-4}$) | 0.4692 ($\pm$0.0228) | 0.3971 ($\pm$0.005) |



Generally, the results show that at maximum filler content (6 wt.%) electrical conductivity increases almost 7–8 decades for the two-component systems with GNP and MWCNT compared with pure PLA, reaching the values of $8.4 \times 10^{-3}$ (S/m) and $2.1 \times 10^{-2}$ (S/m), respectively. Mono-filler systems on the base of MWCNT and PLA show higher values for electrical conductivity, especially at 3 wt.%, where a percolation threshold is reached. On the other hand, the bi-filler composites with PLA/1.5 wt.% GNP/4.5 wt.% MWCNT and PLA/3 wt.% GNP/3 wt.% MWCNT shows a synergetic effect of both nanocarbon fillers on electrical conductivity with the values higher than those observed for the maximum filler content of 6 wt.% of the mono-filler systems. The synergistic effects were attributed to the more efficient MWCNT network, resulting from the presence of GNPs, leading to a more homogeneous filler dispersion. Numerical studies focused on the electrical and thermal transport properties of hybrid nanocomposites indicated that synergy may arise from the presence of the flexible MWCNT rods, which favor the bridging of planar GNP nanoplatelets and thus facilitating the formation of a 3D network [39]. It seems that a combination of conductive carbon fillers (MWCNTs and GNP) enhanced electron transport across the junctions of GNP sheets and tubes, and thus a significant increase in the electrical conductivity of the composites was achieved and proved by the structural analysis. The morphological analysis carried out (Figure 2a,b) indicated that the hybrid bi-filler composites incorporating a small amount of GNPs exhibit quite well-distributed small nanoplatelet aggregates, indicating that CNTs can prevent GNP aggregation and thus improve their dispersion in the PLA matrix. The strong *p-p* interactions and large Van der Waals forces between GNPs can cause their aggregation and stacking in nanocomposites. However, the long MWCNTs acting like arms and bridging GNPs may inhibit their aggregation, thus creating a strong inter-connected hybrid nanofiller network in the polymer matrix [40], as described in the recent literature for thermoplastic and thermosetting matrices incorporating different hybrid fillers [41–43].

Hybrid systems also exhibited improved nanomechanical properties after nanoindentation tests as shown in our previous publication [44]. Synergistic effects were observed in GNP/MWCNT/PLA hybrid composites, when combining GNPs and CNTs at a ratio of 3% GNP/3% CNT and 1.5% GNP: 4.5% CNT. The hybrid filler composites have shown a higher Young's modulus of elasticity and hardness properties with respect to the mono-filler systems incorporating individual CNTs and GNPs at the same overall filler concentration. This improvement was attributed to a synergetic combination of CNTs and GNPs, hypothesizing that the long tortuous CNTs limit graphene aggregation and bridge adjacent graphene platelets creating a more efficient network that has better reinforcing effects in the PLA matrix. This means that the application of MWCNTs as a nanofiller—with geometry different to the layered structure of graphene nanoplatelets—has great effect on the mechanical properties of rubber composites.

### 3.3. Effect of Filler Concentration and Combination of Nanofillers on the THERMAL Conductivity

Generally, the thermal conductivity is a measure of the ability of a material to transfer heat. Thermal diffusivity (with the unit $mm^2/s$) is a material-specific property for characterizing unsteady heat conduction. This value describes how quickly a material reacts to a change in temperature. Thermal diffusivity is the thermal conductivity divided by density and specific heat capacity at constant pressure. It measures the rate of transfer of heat of material from the hot side to the cold side. Thermal management is a crucial issue in the electronics industry as electronic devices are more integrated and miniaturized. If the heat is not dissipated promptly, the efficiency and the lifetime of the system could be reduced, or even damaged. Thus, a material with high thermal conductivity is needed to dissipate the heat [14]. Thermal conductivity through a polymer is a complex process, influenced by many parameters like temperature, crystallinity, orientation of the macromolecules, and so on [45–47]. Phonons are usually considered to be thermal carriers in polymers because there is a free electron [48].

In our study, GNP and MWCNTs are used as the fillers to enhance the thermal conductivity of PLA polymer materials. Figure 6a and Table 3 presents thermal conductivity vs. GNP and

MWCNT filler content for mono-filler (PLA/GNP and PLA/MWCNT) and bi-filler (PLA/GNP/MWCNT) composites.

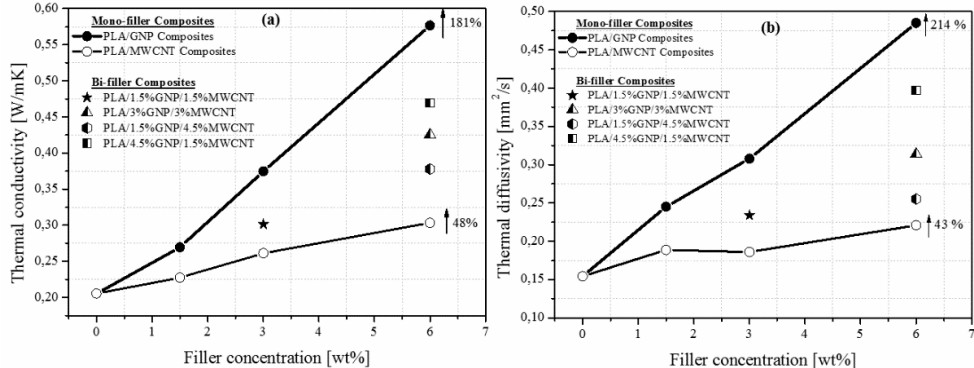

**Figure 6.** (**a**,**b**)Thermal conductivity and thermal diffusivity vs. GNP and MWCNT filler content for two (PLA/GNP and PLA/MWCNT) and bi-filler (PLA/GNP/MWCNT) composites.

The results show that the thermal conductivity or the ability of a material to transfer heat increases almost linearly with increasing of the filler content. This effect was more pronounced for the mono-filler composites based on PLA/GNP where the maximum value is 0.577 (W/mK) at PLA/6 wt.% GNP compared with the 0.205 (W/mK) for the pure PLA. The improvement of thermal conductivity for these systems is about 181% compared with the neat matrix. This can be attributed to the high thermal conductivity of the graphene when compared to the pristine polymeric matrix and the networks created by contact between the graphite layers with increasing of the filler content, thus forming a thermal conduction pathway. The improvement in thermal conductivity, can be related to the interfaces between fillers and matrix. A better adhesion between fillers and host material can suppress phonon scattering, leading to higher values of thermal conductivity. The effect of increasing of the filler content on the thermal conductivity of the mono-filler composites based on PLA/MWCNT was less pronounced and at a maximum MWCNT filler content of 6 wt.% the improvement was of about 48%, due to the shape of the carbon nanotubes and hence poor thermal conductive network of the filler. The thermal conductivity of graphene and carbon nanotubes is mainly accomplished by phonons and the thermal conduction through a polymer is a complicated process, which is influenced by many parameters like crystallinity, temperature, orientation of the macromolecules. On the other hand, carbon nanofillers and polymers are usually considered to have lots of defects that contribute to numerous phonon scatting, leading to low thermal conductivity. MWCNT used in this study had a lower thermal conductivity, compared with GNP. In the mixed bi-filler systems, thermal conductivity depends mainly on the amount of GNP, which is a lower quantity of GNP, compared with the mono-filler GNP/PLA systems, at the same total filler content. This was the reason for no presence of synergetic effect with the combination of both fillers, in comparison with electrical conductivity, where both nanofillers showed high electrical conductivity.

Figure 6b presents thermal diffusivity vs. GNP and MWCNT filler content for mono-filler (PLA/GNP and PLA/MWCNT) and bi-filler (PLA/GNP/MWCNT) composites. The results showed that the thermal diffusivity (the rate of transfer of heat) increases with increasing of the filler content and this effect is more pronounced for the mono-filler composites based on PLA/GNP, where the maximum value is 0.485 (mm$^2$/s) at PLA/6 wt.% GNP compared with the 0.154 (mm$^2$/s) for the pure PLA. The improvement of thermal diffusivity for these systems was about 214%, compared with the neat matrix due to the sheet structure of GNP filler. The effect of increasing of the filler content on the thermal diffusivity of the mono-filler composites based on PLA/MWCNT was less pronounced and at a maximum MWCNT filler content of 6 wt.% the improvement is of about 43%. Thus, obtained results open potential applications for graphene-enabled thermal management, including electronics, which could greatly benefit from graphene's ability to dissipate heat and optimize electronic devices.

In micro-and nano-electronics, very often the heat is a limiting factor for smaller and more efficient components. Therefore, graphene or its composites with exceptional thermal conductivity hold enormous potential for this kind of applications. The existence of synergy between different types of carbon nanofillers, shows great potential and could significantly increase applications of carbon-based nanomaterials. This bi-filler approach can be suitable also for stereo-lithography 3-D printing and this will be a topic of our interest for further study.

Thus, obtained composite materials were extruded into 1.75 mm diameter filaments, which were applicable to commercial FDM 3D printers. In our previous work [49], on the base of rheological characteristics as percolation threshold and the flow index, as obtained nanocomposites were classified into three groups: Newtonian, percolated composites and elastic solids. As a result, these two characteristics were used to select the printing parameters for the three groups of nanocomposites, suitable for FDM. In order to achieve a good accuracy for the printing objects, the printing of the neat PLA and the nanocomposite systems with GNP and MWCNTs was accomplished at appropriate conditions as constant volumetric flow rate for the neat polymer and the composites, and the printing speeds were increased in correlation with the flow index of the printed material. All these characteristics were determined for the fixed melt temperature and nozzle diameter. This concept was validated by direct printing of samples and, using the filament from the neat PLA, the MWCNT/PLA and GNP/PLA nanocomposite formulations studied. Figure 7 summarizes a scheme for obtaining mono-and bi-filler filaments for a FDM 3D printer on the base of PLA, GNP and MWCNTs.

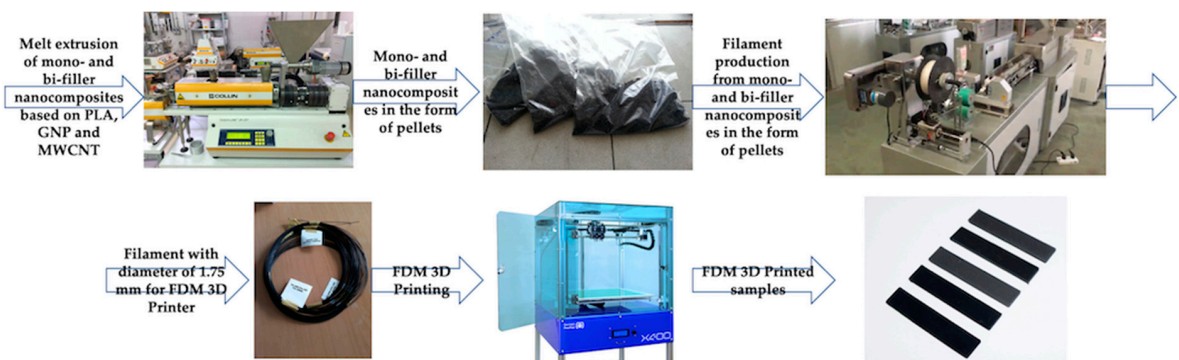

**Figure 7.** Process scheme for obtaining mono-and bi-filler filaments for a FDM 3D printer on the base of PLA, GNP and MWCNTs.

In our previous studies [50–52] it was found that the PLA is still biodegradable and, only if there is a burning of composites (T < 850 °C), residual ash contains an amount of graphite and carbon nanotubes, creating concerns about environmental pollution and damage. A quantitative and qualitative risk assessment has been carried out in two cases—so-called "bad practices" (not in compliance with safety measures) and "good practices" (subject to modern safety measures). It has been shown that the implementation of specific safety measures can significantly reduce the degree of risk.

## 4. Conclusions

The electrical, thermal and morphological properties of PLA/MWCNT/GNP nanocomposites prepared with different contents of graphene and MWCNT were investigated and discussed in this work. The results show that at maximum filler content (6 wt.%) electrical conductivity increases with almost 7–8 decades for the mono-filler systems with GNP and MWCNT compared with pure PLA, but this effect is more pronounced for PLA/MWCNT composites. On the other hand, some of the bi-filler composites with PLA/MWCNT/GNP show a synergetic effect on electrical conductivity with the values higher than those obtained for the maximum filler content of 6 wt.% for the mono-filler systems. The results indicate a close relationship between the electrical and thermal properties with

the morphology of the as-prepared nanocomposites. As the graphene or MWCNTs loading kept increasing, a continuous and denser network was formed in the polymer matrix. Thermal conductivity and diffusivity of GNP nanocomposites were improved by the addition of graphene nanoplatelets by 181% and 214%, respectively. Thermal transport in obtained PLA/GNP composites is a thriving area of research thanks to graphene's extraordinary heat conductivity properties and its potential for use in thermal management applications. The obtained composites can be considered an excellent electrical and heat conductor for a variety of applications.

**Author Contributions:** The study was conceived and designed by E.I. and R.K.; E.I., R.D.M., C.S. and S.C. were involved in melt extrusion of all samples and SEM and TEM analysis; E.I., H.X. and Y.C. performed electrical and thermal conductivity measurements and analysis; V.A. was involved in nanocomposites samples preparation; R.K.D., K.D. and A.P.G. made and wrote the Raman part of investigations; E.I. wrote the paper.

**Funding:** This work has received funding from the European Union's Horizon 2020-MSCA-RISE-734164 Graphene 3D Project.

**Acknowledgments:** Authors from OLEM and NanoTech Lab are grateful to the support from H2020-SGA-FET-GRAPHENE-2017-785219 Graphene Core 2. The author would like to acknowledge the contribution of the COST Action CA15107. Support for the COST CA15107—contract DKOST 01/7, 20.06.2017 with NSF-MER of Bulgaria is also acknowledged. This work is also supported by the National Key R&D Program of China (2017YFE0111500). R.K. Donato, K.Z. Donato and A.P. Godoy acknowledge the financial support of MackPesquisa (Project number 181009). Authors are grateful to G. Rollo, IPCB-CNR, Pozzuoli for the support during the processing of the composites and to P. Lamberti, G. Spinelli and V. Tucci, University of Salerno for the robust design pre-planning.

**Conflicts of Interest:** The authors declare no conflict of interest.

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
