# Peer review of "PLA/Graphene/MWCNT Composites with Improved Electrical and Thermal Properties Suitable for FDM 3D Printing Applications"

_applsci, doi:10.3390/app9061209_

Round 1

Reviewer 1 Report

My main concerns were addressed by the authors as follows:

a) The authors addressed my first main concern, that there is too little data and the results are not representative by adding new results in Table 3. The synergetic effect can be confirmed, again only for high concentration (6%), not for low concentration. Also more position resolved data was added. I appreciates these extra measurements, and consider this question as resolved.

b) The synergy effect was attributed to a more homogenuous network formation. Considering the images from Fig. 2 I believe this is a plausible explanation. However, this cannot be considered to be concluded. E.g. the question why at lower concentration the synergy effect does not work is not clarified. This is however acceptable for this publication. The explanation on why the thermal conductivity does not follow the same trend was also addressed, however not fully resolved.

My other concerns were addressed as follows:

-Biodegradability was commented on, but not in the manuscript. I recommend doing so, also if biodegradability is questionable.

My question whether this concern can be used for optical lithography was responded by restating that this paper was on melt extrusion. I am aware of this method, however I would recommend to make reference also to other disciplines - also in the interest of visibility of this paper. Also for optical methods fillers were used to increase conductivity, so a reference would be appropriate.

My question on "How do you make sure, that you do not measure the silver coating but the bulk material?"

Was answered as follows:

"Polymers have electrical resistivities that are characteristically very high and in most electrical applications they are used as insulators. Thus, antistatic treatments are to surfaces to prevent unwanted accumulation of charge. If we coat specimen with thin layer of silver it will provide a path for these electrons to ground and charging will be prevented."

This does not answer my question. There is a misunderstanding. I assumed that you measure conductivity with a silver coating on your material. Correct? At least you state ""A Picoammeter (Keithley 2400, Keithley Instruments Inc, USA) was used to measure the 119 electrical conductivity of the mono- and bi-filler polymer composites. Before measurements the 120 samples were coated with silver coating. During the electrical measurements, the resistance of the 121 material in ohm was obtained. Electrical conductivity was calculated using the following equation"

 Would this not change your measurements? Please comment in the method and explain why silver coating is no problem.

My question about coating for SEM images "Why do you cover your samples with metal for SEM? If you have could conductivity, this would not be required." was answered by a general explanation about why a coating for SEM images are required.

That was not my question. I was asking why the coating is still required when the sample is conducting, at least for high filler concentration. If you additionally coat your samples, that causes the suspicion that the sample is not conductive. Please comment in the manuscript on why you used a coating on conductive samples. A possible explanation would be, that the coating is required for low filler content, and to have comparable situations you coat all samples. Besides: Using low voltage (800V-1.5kV) measurements without coating can be easily performed on polymer, also with the resolution you show in your paper. The trick is, that the electron column is still operated at 12kV or more and just before hitting the sample the electrons are decelerated to 1kV.

Concerning the specs of your grating: Can you PLEASE correctly use "grating" instead of "grading" and also fix the unites to grooves/mm.

My question on Raman spectra "Fig. 3: Indicate whether these measurements are representative, e.g. do you always achieve the same results?" are answered as follows:

Answer: "The Raman results represent an average of 25 Raman single spectra for each sample."

Are they measured at different positions? Please comment in the manuscript.

Author Response

Responsesto Reviewer 1

Comments and Suggestions for Authors

My main concerns were addressed by the authors as follows:

Point 1: a) The authors addressed my first main concern, that there is too little data and the results are not representative by adding new results in Table 3. The synergetic effect can be confirmed, again only for high concentration (6%), not for low concentration. Also more position resolved data was added. I appreciates these extra measurements, and consider this question as resolved.

Response 1: Thank you very much!

Point 2: b) The synergy effect was attributed to a more homogenuous network formation. Considering the images from Fig. 2 I believe this is a plausible explanation. However, this cannot be considered to be concluded. E.g. the question why at lower concentration the synergy effect does not work is not clarified. This is however acceptable for this publication. The explanation on why the thermal conductivity does not follow the same trend was also addressed, however not fully resolved.

Response 2: Thank you very much for you question. The appropriate text was added to the manuscript, highlighted in yellow:

“MWCNT used in this study has a lower thermal conductivity, compared with GNP. In the mixed bi-filler systems, thermal conductivity depends mainly on the amount of GNP, which is lower quantity of GNP, compared with the mono-filler GNP/PLA systems, at the same total filler content.”

My other concerns were addressed as follows:

Point 3: -Biodegradability was commented on, but not in the manuscript. I recommend doing so, also if biodegradability is questionable.

Response 3: Thank you very much for you question. The appropriate text and references were added to the manuscript, highlighted in yellow:

“In our previous studies [51-53] it was found that the PLA is still biodegradable and only if there is a burning of composites (T <850 ° C), residual ash contains amount of graphite and carbon nanotubes, creating concerns about environmental pollution and damage. A quantitative and qualitative risk assessment has been carried out in two cases - so-called "bad practices" (not in compliance with safety measures) and "good practices" (subject to modern safety measures). It has been shown that the implementation of specific safety measures can significantly reduce the degree of risk.”

51.       Kotsilkov, S.; Ivanov, E.; Vitanov, N. K. Release of Graphene and Carbon Nanotubes from Biodegradable Poly(Lactic Acid) Films during Degradation and Combustion: Risk Associated with the End-of-Life of Nanocomposite Food Packaging Materials. Materials. 2018, 11 (12), 2346. [CrossReff]

52.       Velichkova, H.; Petrova, I.; Kotsilkov, S.; Ivanov, E.; Vitanov, N. K.; Kotsilkova, R. Influence of polymer swelling and dissolution into food simulants on the release of graphene nanoplates and carbon nanotubes from poly(lactic) acid and polypropylene composite films. J. Appl. Polym. Sci, 2017, 134, 45469. [CrossRef]

53.       Velichkova, H.; Kotsilkov, S.; Ivanov, E.; Kotsilkova, R.; Gyoshev, S.; Stoimenov, N.; Vitanov N. K. Release of carbon nanoparticles of different size and shape from nanocomposite poly (lactic) acid film into food simulants. Food Additives & Contaminants: Part A, 2017, 34 (6), 1072-1085. [CrossRef]

Point 4: My question whether this concern can be used for optical lithography was responded by restating that this paper was on melt extrusion. I am aware of this method, however I would recommend to make reference also to other disciplines - also in the interest of visibility of this paper. Also for optical methods fillers were used to increase conductivity, so a reference would be appropriate.

Response 4: Thank you very much for you question. The appropriate text and references were added to the manuscript, highlighted in yellow: 

“This bi-filler approach can be suitable also for stereo-lithography 3-D printing and this will be a topic of our interest for further study.”

Point 5: My question on "How do you make sure, that you do not measure the silver coating but the bulk material?"

Was answered as follows:

"Polymers have electrical resistivities that are characteristically very high and in most electrical applications they are used as insulators. Thus, antistatic treatments are to surfaces to prevent unwanted accumulation of charge. If we coat specimen with thin layer of silver it will provide a path for these electrons to ground and charging will be prevented."

This does not answer my question. There is a misunderstanding. I assumed that you measure conductivity with a silver coating on your material. Correct? At least you state ""A Picoammeter (Keithley 2400, Keithley Instruments Inc, USA) was used to measure the 119 electrical conductivity of the mono- and bi-filler polymer composites. Before measurements the 120 samples were coated with silver coating. During the electrical measurements, the resistance of the 121 material in ohm was obtained. Electrical conductivity was calculated using the following equation"

Would this not change your measurements? Please comment in the method and explain why silver coating is no problem.

Response 5: Thank you very much for you question. The appropriate text and references were added to the manuscript, highlighted in yellow:

“Silver coating doesn’t influence measurements, since it is measuredthe bulk conductivity while the sample is between two electrodes and silver coating is on the both side of the disc samples (with diameter of 16 mm and thickness 3mm). The edges of the samples are enough thick (3 mm) and they are not covered with the silver painting. That’s why there is no contact between two silver painted surfaces.”

Point 6: My question about coating for SEM images "Why do you cover your samples with metal for SEM? If you have could conductivity, this would not be required." was answered by a general explanation about why a coating for SEM images are required.

That was not my question. I was asking why the coating is still required when the sample is conducting, at least for high filler concentration. If you additionally coat your samples, that causes the suspicion that the sample is not conductive. Please comment in the manuscript on why you used a coating on conductive samples. A possible explanation would be, that the coating is required for low filler content, and to have comparable situations you coat all samples. Besides: Using low voltage (800V-1.5kV) measurements without coating can be easily performed on polymer, also with the resolution you show in your paper. The trick is, that the electron column is still operated at 12kV or more and just before hitting the sample the electrons are decelerated to 1kV.

Response 6: Thank you very much for you question. In our future study we shell try to use the measuring technic suggested by the reviewer for the SEM measurements, but also, we will try to use the BSE detector. We planning to buy new SEM equipment, which have a low voltage option and we can apply the suggested measurement.

Point 7: Concerning the specs of your grating: Can you PLEASE correctly use "grating" instead of "grading" and also fix the unites to grooves/mm.

Response 7: Thank you very much for your suggestions. The "grading" is replaced with "grating" and also the unites are fixed to grooves/mm in the manuscript highlighted in yellow.

Point 8: My question on Raman spectra "Fig. 3: Indicate whether these measurements are representative, e.g. do you always achieve the same results?" are answered as follows:

Response 8: "The Raman results represent an average of 25 Raman single spectra for each sample."

Point 9: Are they measured at different positions? Please comment in the manuscript.

Response 9: Thank you very much for you question. The appropriate text and references were added to the manuscript, highlighted in yellow:

The Raman results represent an average of 25 Raman single spectra for each sample at different positions.

Reviewer 2 Report

The authors have addressed my previous comments and I recommend the paper for publication.

Author Response

Point 1: The authors have addressed my previous comments and I recommend the paper for publication.

Response 1: Thank you very much!

Reviewer 3 Report

Most of the comments that I have addressed are not utilized to really improve the manuscript. There is only minor changes made.

My recommendation is still to reject the submission.

Author Response

Thank you for your comments.

Reviewer 4 Report

It is not clear for me how Raman mapping was done. As it is mention in the paper, mapping was done to evaluate the nanocarbon fillers structure/quality within the polymer, so I understand that the data were also collected from the cross section of the composites (and not only from the surface) or in three dimensions, to yield XY images, XZ and ZY slices.

It should be also shown if all the synthesized composites were able to be extruded into 1.75 mm diammeter, since those with high filler concentration could form carbon agglomerates and hinder extrusion and 3D printing.

Author Response

Responses to Reviewer 4

Comments and Suggestions for Authors

Point 1:It is not clear for me how Raman mapping was done. As it is mention in the paper, mapping was done to evaluate the nanocarbon fillers structure/quality within the polymer, so I understand that the data were also collected from the cross section of the composites (and not only from the surface) or in three dimensions, to yield XY images, XZ and ZY slices.

Response 1:The authors agree that the text of the experimental part concerning Raman spectroscopy was not clear. In fact, single spectra Raman were performed for carbon nanofiller quality control. Raman mapping was performed the selectively evaluate the distribution of GNP and CNT by mapping individually the concentration of the D, G and 2D bands within a longitudinal section on the hot pressed films. Although the analysis is performed directly on the hot pressed film, it is not really surface analysis, since the laser is focused just below the surface and covers a representative area of the film.

            Modifications: The experimental part referring to Raman spectroscopy analysis was modified with a more complete discussion, as described below (Line 147-158):

            The Raman spectra of carbonaceous components were obtained using a Raman Confocal Spectrometer (WITec Alpha 300R) with 50x objective lens, grating of 600 grooves/mm and excitation laser wavelength of 532 nm. Laser beam aperture of 1.2 micrometres and measured laser power of 3 mW. Raman spectroscopy is an efficient technique to characterize graphene in terms of quality, structural integrity and number of layers. Thus, single spectra were collected to evaluate the nanocarbon fillers structure/quality within the pelletized filament, collecting an average of 25 spectra per sample at different locations for excluding localized effects. Moreover, Raman spectral imaging (mapping) was performed on hot-pressed films (at 200°C for 10 min) to selectively evaluate the distribution of GNP and CNT by mapping individually the distribution of the D, G and 2D bands within a longitudinal section of hot-pressed films. Prior to the application of mapping, single spectra of the hot-pressed films were performed to localize the appropriate focus position and detect any effect caused by the hot pressing in the carbonaceous component.

Point 2:It should be also shown if all the synthesized composites were able to be extruded into 1.75 mm diammeter, since those with high filler concentration could form carbon agglomerates and hinder extrusion and 3D printing.

Response 2:Thank you very much for you question. The appropriate text were added to the manuscript (with supporting reference 54), highlighted in yellow [Line 352-362 and Line 543-545]:

“In our previous work [54], on the base of rheological characteristics as percolation threshold and the flow index, as obtained nanocomposites were classified in three groups: Newtonian, percolated composites and elastic solids. As a result, these two characteristics were used to select the printing parameters for the three groups of nanocomposites, suitable for FDM. In order to achieve a good accuracy for the printing objects, the printing of the neat PLA and the nanocomposite systems with GNP and MWCNTs was accomplished at appropriate conditions as constant volumetric flow rate for the neat polymer and the composites, and the printing speed were increased in correlation with the flow index of the printed material. All these characteristics were determined for the fixed melt temperature and nozzle diameter. This concept was validated by direct printing of samples, using the filament from the neat PLA, the MWCNT/PLA and GNP/PLA nanocomposite formulations studied.”

54. Ivanova, R.; Kotsilkova, R. Rheological study of poly(lactic) acid nanocomposites with carbon nanotubes and graphene additives as a tool for materials characterization for 3D printing application. Appl. Rheol, 2018, 28, 54014 [CrossRef]

This manuscript is a resubmission of an earlier submission. The following is a list of the peer review reports and author responses from that submission.

Round 1

Reviewer 1 Report

The authors present an investigation of nano-composites with different contents of two different fillers, namely graphene and multi-walled nanotubes. They also investigate the combination of these fillers.

The authors use SEM/TEM/Raman-microscopy/conductivity measurements and thermal conductivity measurements. The findings are an improvement of thermal and electrical conductivity with increased filler content.  They find a synergy effect for electrical conductivity, not for thermal conductivity. The authors give no plausible explanation for the observation of the synergy effect.

While the publication as such appears to be relevant. Besides minor correction, see below, for publication two aspects are missing:

a)      The investigations are very local, it is unclear whether they are accidental findings or a material property. Please increase your database.

b)     There is no plausible explanation for the synergy effect.

Please expand on these points in particular and on all points listed below – if this is successfully achieved, I recommend publication.

-Explain ALL apprehension, even in your abstract there are unexplained apprehensions.

-Consequences of your findings of bi- and monofiller unclear

-Comment on whether PLA is still biodegradable when filled with composites

-Explain whether and how your approach can be used for optical lithography. If you can show how a bi-filler approach allows using composites for two-photon polymerization or similar technologies that would be very interesting for these communities.

-112: How do you make sure, that you do not measure the silver coating but the bulk material?

-123: how do you make sure that heat capacitance plays no role?

-Very local probing methods like SEM and TEM are not suitable to confirm dispersion inside a matrix. Please extend your experiments or explain how you can be sure that you have achieved a homogenous matrix

-Why do you cover your samples with metal for SEM? If you have could conductivity, this would not be required.

-134 What is a “grading 600 g/mm”?

-Fig. 1: I do not understand what I see here. Please label. In particular I cannot see any fine dispersion of nanoparticles.

-Fig. 1: Comment on whether this finding is representative e.g. if you do the same experiment somewhere else, do you observe the same findings?

-Fig. 1: Can you supplement these images with another technique showing the homogeneous dispersion over a larger area? E.g. with dark field microscopy or another suitable method

-Fig. 3: Indicate whether these measurements are representative, e.g. do you always achieve the same results?

-Fig. 4: To my understanding, the top left image indicates, that nanoparticles are not well dispersed. Think it is very important to expand on these finding by analysing larger areas or more sample positions.

-Fig. 5: The key finding of the paper is that the half-filled triangle and octagon are above the filled/not-filled circle. All other data points are trivial. Please present your findings more clearly.

-Please repeat the measurement for 6  wt. % filler for at least 20 position each. This would indicate the variance of the measurement and give a good idea on how well the dispersion of the nanocomposites was achieved.

-The explanation of the synergy effect for electric conductivity is not precise – be more accurate here. The explanation you give “This [synergy] effect is related to the structure and particle geometry of the hybrid fillers, the interactions between the fillers, the concentrations and the processing methods as the synergy refers to the interaction of elements whose combination produces a total effect that is greater than the sum of the effects of the individual elements. It seems that a combination of conductive carbon fillers (MWCNTs and GNP) by which electron transport was enhanced across the junctions of GNP sheets and tubes and a significant increase in the electrical conductivity of the composites was achieved and proved by the structural analysis.” is equivalent to saying that you do not know – While it should be appreciated to be honest on this point, I recoment further investigation by either experiment or relevant theoretical models.

-You do not explain why there is no synergy effect for thermal conductivity while there is for electrical conductivity. Please do so.

-Generally I would recommend not to slice results. This work could have been combined with “Nanoindentation analysis of 3D printed poly(lactic acid)‐based composites reinforced with graphene and multiwall carbon nanotubes”, the overall picture would be more helpful and more relevant than two disjunct publications

Reviewer 2 Report

This paper studies different nanofiller materials and concentrations to make PLA conductive. Such 3D printable composite materials are a timely subject and this paper makes an interesting contribution to the field. I therefore recommend it for publication. I have the following comments that the authors should address:

- Line 51: There should be no comma after nanomaterials.

- The trends observed for electrical and thermal conductivity are different. Electrical conductivity is higher for the MWCNT composite compared with GNP and there is a synergistic effect. Thermal conductivity is higher for GNP and there is no synergistic effect. The authors should comment on possible reasons for this.

Reviewer 3 Report

The paper seems not contain enough new and significant scientific information adequate to justify publication. 

Topic and whole article is misleading since it seems that no real 3D printing test were done.

The authors has just mixed some materials and make filaments from those, which might be 3D printable but NOT tested at all.

The reporting of the study is poor. The reporting of the experimental methods and results should be more complete and accurate.

In AM I recommend to use ISO / ASTM52900 – 15 Standard Terminology for Additive Manufacturing – General Principles – Terminology. For the sake of clarity and for future understandability and indexing when standard name overrules other.

The Abstract would benefit from quantitate data about the results.

The test parts made with parameters and measurements should be gathered in table format. Now text is quite hard to read.

About the design of the sample? How those look?

About  Mechanical characterization. How many samples there were? How much variation there was between measurements? What was the repeatability?

Explain in detail how the experiments of the samples have been planned and analyzed (factors, levels, type of experimental plan, replications, analysis of variance and related statistical tests. Was there predefined experimental design, e.g. a factorial plan? Explain how the process variables(if tested) influence the responses (individual effects and possible interactions).

I recommend reject the submission.